# Influence of Resistance Training on Gait & Balance Parameters in Older Adults: A Systematic Review

**DOI:** 10.3390/ijerph18041759

**Published:** 2021-02-11

**Authors:** Christopher J. Keating, José Carlos Cabrera-Linares, Juan A. Párraga-Montilla, Pedro A. Latorre-Román, Rafael Moreno del Castillo, Felipe García-Pinillos

**Affiliations:** 1Department of Didactics of Music, Plastic and Corporal Expression, University of Jaén, 23071 Jaén, Spain; jccabrer@ujaen.es (J.C.C.-L.); jparraga@ujaen.es (J.A.P.-M.); platorre@ujaen.es (P.A.L.-R.); rmoreno@ujaen.es (R.M.d.C.); 2Department of Physical Education and Sport, University of Granada, 18011 Granada, Spain; fgpinillos@ugr.es; 3Department of Physical Education, Sport and Recreation, Universidad de La Frontera, Temuco 480011, Chile

**Keywords:** resistance training, strength training, aging, gait, balance, walking speed

## Abstract

In this work we aimed to perform a systematic review of randomized controlled trials within an aging population that investigated the general impacts of a resistance training (RT) protocol on key outcome measures relating to gait and/or balance. Following the Preferred Reporting Items for Systematic Reviews and Meta-analyses (PRISMA) statement guidelines, two electronic databases (PubMed, and Scopus) were searched for randomized controlled trials that measured at least one key outcome measure focusing on gait and/or balance in older adults. 3794 studies were identified, and after duplicates were removed, 1913 studies remained. 1886 records were removed due to the abstract not meeting the inclusion criteria. 28 full-text articles were assessed further, and 20 of the articles were identified as meeting the criteria for inclusion. The remaining 20 studies were assessed for quality using the Physiotherapy Evidence Database (PEDro) scale; 12 studies remained and were included in this systematic review. Our review suggests that RT has a positive effect on both gait and balance in an elderly population. RT improves gait, specifically straight-line walking speed in older adults. RT is an adequate training method to improve balance in an aging population. Improvements in strength, attributed to RT, may allow for greater autonomy and independence to carry out activities of daily living as we age.

## 1. Introduction

The world’s population is aging and it is creating a unique situation in which the population over 65 years of age exceeds that of children under 5 years of age [1]. Currently, 11% of the world population is over 60 years of age. The population aging trend continues and it is projected that by 2050 this population will include more than 22% of the world population [2]. In light of these calculations, active aging is presented as one of the best options to allow the elderly to enjoy a higher quality of life and a higher level of health to be the protagonists of their own lives in advanced age. By doing so they can avoid spending excessive life years and money on costly medical care and treatment [3].

Physical activity (PA) is presented as an alternative to medicine in terms of improving quality of life since it has been proven to have positive physiological effects in an aging population (i.e., prevents chronic diseases and reduces the risk of non-communicable diseases) [4]. In this sense, the lack of physical activity is what causes the adverse effect, resulting in what is known as frailty. This is a syndrome that appears when 3 or more of the following criteria are present in a person who suffers from it: weight loss, weakness, slowness, exhaustion, and low levels of PA. Therefore, the term frailty encompasses various aspects such as gait, mobility, balance, muscle mass, motor processing, cognition, nutrition, endurance, and PA [5]. In those individuals over the age of 65, frailty causes a greater risk of falling, which is the second cause of death and injury in the world population and it is becoming a serious public health problem for the elderly [6]. One-third of the aging population falls at least once a year, and a fall in an elderly individual can have serious consequences such as life-threatening injury, hospitalization, fractures, and/or a loss of independence [7]. Falling or simply the fear of falling can result in a restriction of physical activity levels, and indirectly in the reduction of social interactions. This causes a paradox in which the fear of falling can increase the risk of future falls due to the deterioration of physical abilities from not participating in everyday life [8].

The physical inactivity derived from a fall can accentuate the loss of muscle mass and strength to a greater extent than that caused alone by age-associated loss. It is often reported that muscle mass decreases by roughly 2% each year after the age of 50 or, similarly, by 15% for every 10 years after the age of 50 [9]. This progressive loss of strength and muscle mass is known as Sarcopenia [10]. The term Dynapenia can also be used to further describe the age-related loss of muscle strength and power that is not caused by neurologic or muscular diseases [11]. Sarcopenia/Dynapenia and frailty cause a progressive deterioration of functional ability that is heightened in older ages. A gait speed greater than 1.20 m/s is associated with greater independence in older adults, while a speed less than 0.8 m/s is a predictor of future dependence that can lead to hospitalization, medical care, cognitive decline, and mortality at these ages [12].

Traditionally, aerobic training programs have been used to reverse the effects of the above-mentioned pathologies, as well as an improvement in the health status of the elderly [13]. This has been shown to improve cardiorespiratory function, decrease hypertension, and improve functional activities (e.g., muscle strength, physical performance, and decrease the risk of falls). In the same way, it can also improve cognitive function, while also having a positive impact on improving quality of life [14]. However, resistance training (RT) is also an appropriate exercise training method to improve health parameters and when used in combination with aerobic exercise it has been shown to improve functional capacity in an aging population [15]. In this regard, resistance training is defined as any exercise that causes the muscle to contract against resistance (weights, bands, external objects, body weight, etc.) with the intention of provoking physiological and/or morphological changes.

Recent pilot data and theoretical reviews have suggested that RT in the elderly could be an effective and safe method of participating in PA that is capable of reversing the effects of sarcopenia [16], as well as an improvement in body posture, balance, and physical resistance [17,18]. Therefore, resistance training must be a key component to be introduced in training programs for the elderly since, in addition to the benefits mentioned, it may produce neuromuscular improvements such as increased muscle mass, strength, and functional capacity [19]. However, a large amount of this information is based upon outdated data sets. A systematic review from 2004 suggested that RT is a promising exercise regimen for older adults but more research was needed to determine its effectiveness [20]. Another systematic review and meta-analysis from 2010 found promising results, but concluded that further research is needed to provide more considerable conclusions regarding the effect that RT has on the functional performance of older adults [21].

Therefore, this work aimed to perform a systematic review of randomized controlled trials within an aging population that investigated the general impacts of a resistance training protocol on key outcome measures relating to gait and/or balance. 

## 2. Methods

This review was conducted following the Preferred Reporting Items for Systematic Reviews and Meta-analyses (PRISMA) statement guidelines [22]. Two electronic databases (PubMed, and Scopus) were searched for randomized controlled trials that measured at least one key outcome measure focusing on gait and/or balance in older adults. Search terms used included: resistance training OR strength training AND balance OR gait. The search terms were limited to TITLE/ABSTRACT/KEYWORDS. The search was further limited to “clinical trials”, in “humans”, published in “the last ten (10) years” (January 2010 to June 2020), “adult: 65+ years” of age, and published in “English”. 

### 2.1. Study Selection—Inclusion Criteria

The inclusion criteria for this systematic review were full-length research articles published in peer-reviewed academic journals in the English language. Only randomized controlled trials published from January 2010 up to June 2020 were eligible. Studies that included participants with a median age of 60+ years. Resistance training interventions that measured at least one variable relating to gait and/or balance were included.

### 2.2. Study Selection—Exclusion Criteria

Abstracts, conference presentations, poster presentations, letters to the editor, books or book chapters, unpublished papers, proposed protocols, validation studies, or retrospective designs were excluded. Studies were also excluded if the participants were taking supplements, or if the average age of participants was ≤60 years. Also, studies that met the inclusion criteria yet later did not achieve a score of 5 or greater on the PEDro scale were also excluded from the review.

## 3. Results

The initial search resulted in 3794 studies; after duplicates were removed, 1913 studies remained, and the abstracts were reviewed for meeting the inclusion criteria. Following the initial screening process, 1886 records were removed due to the abstract not meeting the inclusion criteria. 28 full-text articles were assessed further and 20 of the articles were identified as meeting the criteria for inclusion. The remaining 20 studies were assessed for quality using the Physiotherapy Evidence Database (PEDro) scale, 8 of the studies did not score 5 or greater and were consequently removed. 12 studies remained, and all were included in the systematic review. (Figure 1)

The Physiotherapy Evidence Database (PEDro) scale is an 11-item scale that rates randomized controlled trials from 0 to 10. One item (eligibility criteria) is included in the scale because it influences external validity but not the internal or statistical validity of the trial, thus it is not counted toward the final score. Therefore, the PEDro score is generated from an 11-item scale resulting in a final score of 0 to 10. Seventeen of the twenty studies were scored directly from the Physiotherapy Evidence Database [23]. The remaining three studies were not included in the database and were scored separately by 2 authors (CJK and JCCL); there was full consensus amongst the authors’ scores. (Table 1) 

### Study Characteristics

Twelve studies were included in the review, and all were published in the English language. The randomized controlled trials were conducted in the following countries: USA = 4 [24,25,26,27], Portugal = 2 [28,29], Australia = 1 [30], Brazil = 1 [31], Chile = 1 [32], Japan = 1 [33], Norway = 1 [34], and Spain = 1 [35]. The total number of participants analyzed in all studies was 499 (only including resistance-trained participants). Eleven of the twelve studies had reported the gender of the participants and approximately 60% of them were female (149 males to 304 females). Three studies reported mean ages of ≥65–69.9 years [27,28,33], 6 studies reported mean ages between 70–79.9 years [24,25,26,29,31,32], two studies reported mean ages between 80–89.9 years [30,34], and one study reporting a mean age of >90 years [35].

Nine of the twelve studies recruited participants that were community-dwelling [24,25,26,27,28,31,32,33,34], whereas three studies recruited participants from residential care facilities [29,30,35]. Of those studies that had recruited participants from the community, only two had reported further underlying conditions; Nicklas et al. included participants that were overweight or obese and Sylliaas et al. investigated hip fracture patients [27,34]. Only one study with participants from residential care facilities reported further underlying conditions, and they reported on “frail” nonagenarians [35]. (Table 2)

Resistance training intervention duration ranged greatly from 6 to 32 weeks, with one study reporting data for 6 weeks [25], one study reporting for 10 weeks [33], four studies reporting for 12 weeks [31,32,34,35], two studies reporting for 16 weeks [24,29], one study reporting for 20 weeks [27], one study reporting for 25 weeks [30], one study reporting for 26 weeks [26], and one study reporting for 32 weeks [28]. All twelve studies described the frequency of training in “days/week”; the studies were split evenly with six studies conducting the intervention 2 days/week [25,30,31,32,33,35] and six studies conducting the interventions 3 days/week [24,26,27,28,29,34].

Regarding the number of sets and repetitions used in the RT interventions, the research appears to be relatively diverse. The number of sets used in the interventions included three interventions using 2 sets [24,26,28], six interventions using 3 sets [25,27,31,32,33,34], one study using 2–3 sets [30], and two studies simply using a “varied” use of sets [29,35]. The number of repetitions used per set of exercise in the respective interventions included one study using 6 to 8 [28], two studies using 8 to 12 [33,34], one study using 8 to 15 [25], one study using 10 to 15 [30], one study only using 10 [27], two studies only using 12 [24,26], one study using a fixed 12, 10, and 8 repetitions model [31], and three studies using “varied” repetitions [32,35,36]. 

All studies reported the type of resistance modalities used during the training sessions. Of which, four studies reported using resistance machines [27,28,31,33], two studies utilizing elastic bands [24,35], two studies utilizing both body weight and machines [25,34], one study utilizing high-speed resistance training with free weights [32], one study utilizing a combination of pneumatic machines and balance training [30], one study utilizing a combination of calisthenics and elastic bands [29], and one study utilizing both body weight and free weights [26]. (Table 3)

All twelve studies reported the dropout rates of their respective participants; three of the studies reported that no participants had dropped out of the resistance training group [25,29,32], two studies reported its dropouts but did not provide explanation [24,26], and seven studies had reported the dropout rate of its resistance training participants with explanations [27,28,30,31,33,34,35]. On the other hand, only three studies reported on adverse events related to the resistance exercise intervention [26,27,30]. Of those three studies that reported adverse events, there was a total of fourteen individual events; thirteen of those events were related to musculoskeletal aches and pains and only 1 event was related to a non-injurious fall [30]. (Table 4)

Seven of the twelve studies reported on variables related to balance alone [25,26,28,29,31,32,35], whereas only one study reported on gait alone [24]; the remaining four studies reported on both gait and balance variables [27,30,33,34]. The most common test used to assess balance was the Timed Up and Go (TUG) or the 8 foot Timed Up and GO (8ftTUG) variation; other tests included the single-leg stance, tandem or bilateral stance, as well as the body’s center of oscillation. Tests assessing gait alone included velocity (m/min), step time (seconds), and step length (cm). Tests in the studies that provided measures for both gait and balance were mixed and included assessments such as the Short Physical Performance Battery (SPPB), 10-m walk speed, Functional Reach Test (FRT), Berg Balance Scale (BBS), the center of oscillation, and the 400-m walk test for time (Table 5).

All twelve studies observed a positive effect of the RT intervention in at least one of the studies’ outcome measures; none of the studies reported a negative effect due to the RT intervention. All eleven studies that analyzed balance specified an improvement in either static and/or dynamic balance. All five studies reporting on gait measures reported a positive effect of the RT intervention, and particularly an improvement in gait speed.

## 4. Discussion

The main objective of this work was to examine the general impact that an RT program has on key outcome measures relating to gait and balance. According to the studies included in this review, it is evident that RT has a positive effect on both gait and balance in an elderly population. 

Regarding gait, only five studies were found to investigate gait parameters. All five of those studies used some form of a timed walking test, four of which evaluated the 10-m walking test, whereas the other measured gait as part of the SPPB (3/4-m walking test). This may be due to the common belief that gait speed itself is the best indicator of gait function, which does fall in line with the findings from Guralnik et al. that suggest that gait speed could be the best predictor of frailty and disability in older adults [37]. However, unidirectional walking speed is simply one of the many methods to analyze gait. This sentiment is echoed by M.W. Whittle, who indicates that walking is only one of the many functions of the musculoskeletal system and that we should “broaden our horizons and use the power of the modern measurement systems to study a wide range of other activities” [38]. Although the authors of this review believe that unidirectional gait assessment is an essential measurement, we also suggest that further research needs to include multidirectional and/or double task scenarios to better understand their utility in analyzing gait in older adults.

It is interesting to note that only one study examined the effects of RT on gait parameters alone and they concluded that eight weeks of resistance training improved the measures of velocity and step length; however, there was no significant increase in step time measured in seconds. Those authors also indicated that it could be possible to see additional gains if an emphasis were placed specifically on gait training and that it is necessary to design programs with a specific objective centered on the target population and/or individual rather than a standardized or “one size fits all” model [24]. The other four studies analyzing gait measured the time of a 10-m walking test, and all of them found significant improvements from baseline to post RT intervention. 

According to the findings included in this review, resistance training undoubtedly improves gait parameters in older adults, but specifically unidirectional walking speed. It is interesting to note that there are other forms of gait parameter tests that are not simply straight-line walking tests [39]. The authors suggest that more research needs to be done on the effects of an RT program on a complex gait or a dual-task scenario. Research has found an association between gait variables and cognitive function in older adults [39]. In this regard, a complex gait test when measuring the time to completion would allow researchers to get a better understanding of the relationship between the functional and cognitive state of the individual. Furthermore, a complex gait test would be a more accurate representation of a real-life scenario, and therefore a better predictor of future adverse events. However, irrespective of straight-line walking speed, more research is needed to determine if RT can enhance the various aspects of gait in older adults.

Regarding balance, 11 studies analyzed at least one balance variable and all of them reported that RT had a significant effect on improving balance; only 1 of the studies analyzed advised concern regarding the improvements from the RT group. That study, by Alfieri et al., could not determine which of the programs included in their research (a multisensory or RT intervention) was more suitable for improving balance control [31]. They further state that although there was no significant between-group difference, the multisensory group showed better improvements in the dorsiflexor and plantar flexor muscles of the ankle which have been demonstrated to be important for the maintenance of static balance. In any case, RT did induce a significant change in measures such as TUG, BBS, and the body’s center of oscillation. 

Numerous variables need to be controlled and/or modified to achieve the desired objectives of improving balance. For that very reason, RT can be difficult to program and prescribe to such a diverse population base [40,41]. Considering that there are many variables requiring attention to develop an effective RT program, it is promising to report that all studies included in this review obtained significant improvements in balance across a wide variety of RT programs. 

The duration of the interventions varied widely from 6 to 32 weeks, with 12 weeks being the most common. It is important to highlight that Gonzalez et al., obtained improvements in balance with a basic RT program consisting of 2 days/week for 6 weeks. This indicates that an RT program with a specific objective (in this case, improved balance) can achieve significant improvements in a relatively short intervention time. This reduction in intervention time could prevent the abandonment of the program by participants, since lack of adherence due to interest is one of the main reasons why subjects cease training [42,43]. This short training time could allow the exercise specialist to include well-deserved breaks for the participants within the macro/mesocycle, as well as changing the program accordingly to make it more desirable for the participants. 

Regarding the number of sets used (2–3) and the number of repetitions (between 8–15), 11 of the articles analyzed used a methodology following the American College of Sports Medicine Position Stand on Progression Models in Resistance Training for Healthy Adults in order to increase muscle mass through hypertrophy [44]. Five of the twelve studies used the 1-repetition maximum (1-RM) method to prescribe training loads [21,22,26,27,28]. Despite its widespread use, this method has some disadvantages that must be considered. Among others, this can be unsafe and harmful for the performer when the subject does not have prior training and/or their performance technique is not correct [45]. The intense efforts of a 1-RM may produce unnecessary musculoskeletal loading that may not be recommended for certain populations such as the elderly. For this population, an alternative method would be to include one of the many 1-RM prediction equations which have been shown to be a good predictor of an individual’s true 1-RM [46].

Several studies analyzed in this review included a variety of rest times between sets from 1 min, 2 min, and 3 min. However, many of the studies analyzed did not report the rest time between sets. The rest time between sets is an important variable to consider when planning an RT program and, surprisingly, more studies did not plan or at least report their rest times appropriately [47]. In this regard, a shorter rest time between sets implies a reduction in the total training time, and therefore the perception of fatigue could also be perceived as less. Common knowledge amongst the scientific literature suggests that the rest time between sets should range between 180–300 s when the objective is to increase maximum strength, 1–2 min for muscle gain through hypertrophy, and 30–60 s for improving muscular endurance [48]. However, a recent study by Villanueva et al. in older adult males concluded that a 60-s rest between sets was optimal for hypertrophic muscular gains, which appear to compensate for the effects produced by age [49]. The difference in rest time between sets, as well as the absence of it in several studies analyzed in this review, makes it difficult to determine to what extent this variable may have influenced the improvement of balance and/or gait variables. Further studies need to be very clear in not only the number of repetitions and sets but further into the rest time between sets as well as total exercise time. 

Due to the differences in the training programs, evaluation methods, and the subject population used in the studies of the current review, it has not been possible for the authors to determine to what extent the variables in these programs has had a greater influence on improving balance and gait. However, it is noteworthy to report that a recent systematic review looking at the effects of supervised vs. unsupervised training programs on balance and muscle strength in older adults suggests that supervised training improved measures of balance and muscle strength/power to a greater extent than that of unsupervised programs [50]. Therefore, the authors of the current review suggest that future studies need to be carried out to focus on the RT variable/s that allow for superior improvements in gait and balance. 

This review also helps identify the feasibility and safety of implementing an RT program in an aging population. Remember that many of the participants from the studies included in this review were over 70 years of age, and although there were a significant number of dropouts reported, the authors did not relate those dropouts to the RT program. Only half of the studies reported adverse events in their respective studies. When adverse events were reported, most of those events (13 of 14) were related to musculoskeletal aches and pains. This is not out of the ordinary at any age, but may be the leading cause of adverse events in an aging population, as stated in a systematic review by Lui and Latham (2010). However, in that same review they report that many adverse events may go undocumented because there is no consensus on reporting, nor the definition of an adverse event. They further state that reporting adverse events in an aging population needs to become part of the standard research protocol to further guide practitioners and further develop research [42]. We would like to echo that opinion and encourage researchers to become more prudent in reporting participants’ adverse events; which could be a simple comment or complaint of simple aches and pains that may arise in day to day conversation with the participants.

Considering the results provided by the different studies analyzed in this review, RT is an adequate method to improve balance in people over 65 years of age. Even in the study in which improvements in balance were questioned, there were still significant improvements in lower body strength in the participants. These improvements in strength can, in turn, lead to greater independence and autonomy to carry out the activities of daily living [51,52].

## 5. Conclusions

This work aimed to review the general impact that an RT program has on key measures relating to gait and balance in older adults. With the studies included in this review, RT has a positive influence on both gait and balance in an aging population. RT enhances gait parameters, but specifically straight-line walking speed, in older adults. It appears that the improvement can be highly attributed to the significant improvements in lower body strength. Nonetheless, it appears that RT is an adequate and safe method to improve balance and gait parameters in people over 65 years of age. However, more research is needed to determine if RT can improve the various and complex aspects of gait in older adults. Furthermore, adverse events often go unreported and should become part of the standard research protocol when partaking in studies on older adults.

## Figures and Tables

**Figure 1 ijerph-18-01759-f001:**
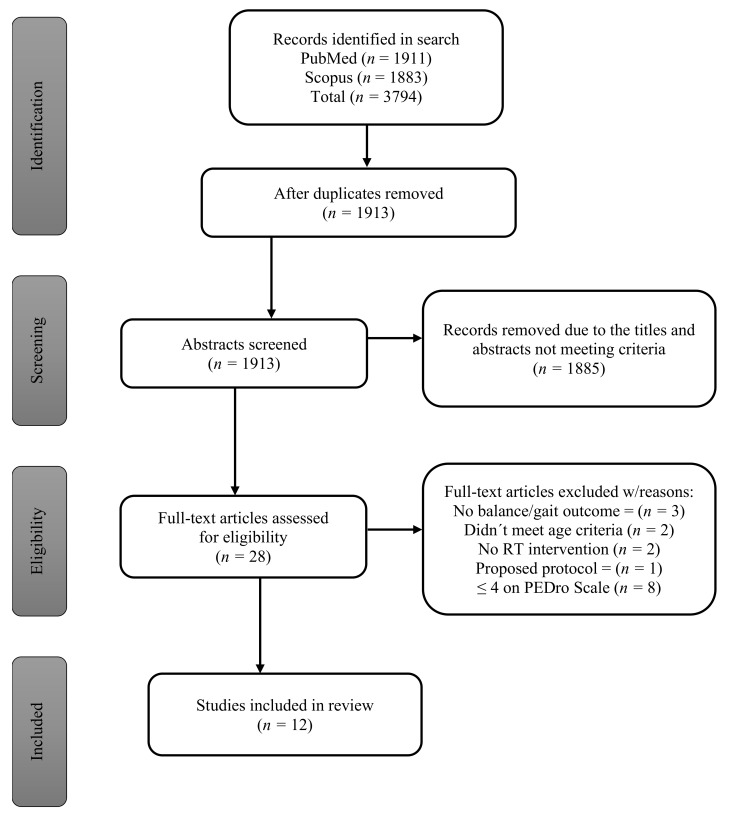
Article selection flow-chart.

**Table 1 ijerph-18-01759-t001:** PEDro—Quality Assessment.

Authors	1 *	2	3	4	5	6	7	8	9	10	11	Total Score
Alfieri et al., (2012)	1	1	0	1	0	0	0	0	1	1	1	5
Cancela Carral et al., (2019)	1	1	0	1	0	0	0	1	0	1	1	5
De Sousa et al., (2013) ▪	1	1	0	0	0	0	0	0	0	1	1	**3**
Fahlman et al., (2011)	1	1	0	1	0	0	1	1	0	1	1	6
Forte et al., (2013)	1	1	0	0	0	0	0	0	0	1	1	**3**
Gonzalez et al., (2014) ▪	1	1	0	1	0	0	0	1	1	1	1	6
Hewitt et al., 2018	1	1	1	1	0	0	1	1	1	1	1	8
Iuliano et al., (2015)	0	1	0	1	0	0	0	0	0	1	1	**4**
Marques et al., (2011)	1	1	1	1	0	0	1	0	1	1	1	7
Martins R, et al. (2011)	1	1	0	1	0	0	0	1	0	1	1	5
Nicholson et al., (2015)	1	1	0	1	0	0	0	0	0	1	1	**4**
Nicklas et al., (2016) ▪	1	1	1	1	0	0	1	1	1	1	1	8
Pamukoff et al., 2014	1	1	0	1	0	0	0	0	0	1	1	**4**
Ramirez-Campillo, Rodrigo, et al. (2016)	0	1	0	1	0	0	1	1	0	1	1	6
Roma et al., (2013)	1	1	0	1	0	0	0	0	0	1	1	**4**
Sahin et al., (2018)	1	1	0	1	0	0	0	0	0	1	1	**4**
Shiotsu & Yanagita, (2018)	1	1	0	1	0	0	0	1	0	1	1	5
Sparrow et al., (2011)	1	1	0	1	0	0	1	1	1	1	1	7
Sylliaas et al., (2011)	1	1	1	1	0	0	1	1	1	1	1	8
Yoon et al., (2018)	0	1	0	1	0	0	0	0	0	1	1	**4**

* Not counted toward total score; ▪ Scored by reviewers; Bolded Total Score ≤4 and therefore not included in this review.

**Table 2 ijerph-18-01759-t002:** Participant characteristics.

Authors	Population	Population (Cont.)	Age	*n* =	Male	Female
Alfieri et al., (2012)	Community-dwelling		70.18 ± 4.8	23	1	22
Cancela Carral et al., (2019)	Residential care	Frail	90.8 ± 4.02	13	0	13
Fahlman et al., (2011)	Community-dwelling		74.8 ± 1	46	NR	NR
Gonzalez et al., (2014)	Community-dwelling		71.1 ± 6.1	23	12	11
Hewitt et al., 2018	Residential care		86 ± 7	113	42	71
Marques et al., (2011)	Community-dwelling		67.3 ± 5.2	23	0	23
Martins et al. (2011)	Residential care		73.4 ± 6.4	23	10	13
Nicklas et al., (2016)	Community-dwelling	Overweight/Obese	69.4 ± 3.6	63	34	29
Ramirez-Campillo et al., (2016)	Community-dwelling		70 ± 6.9	8	0	8
Shiotsu & Yanagita, (2018)	Community-dwelling		69.0 ± 4.1	12	0	12
Sparrow et al., (2011)	Community-dwelling	Vets & Spouses *	70.3 ± 7.5	52	35	17
Sylliaas et al., (2011)	Community-dwelling	Hip Fracture	82.1 ± 6.5	100	15	85

* US Military Veterans and Spouses, NR = not reported.

**Table 3 ijerph-18-01759-t003:** Resistance training intervention details.

Authors	Exercise Modality	Days/Week	Weeks	Sets	Reps	Rest-time	Load	Total Time
Alfieri et al., (2012)	Machines	2	12	3	12,10,8	NR	50%, 75%, MTL	60
Cancela Carral et al., (2019)	Elastic Bands	2	12	varied	varied	30–60 sec	progressive	60
Fahlman et al., (2011)	Elastic Bands	3	16	2	12	NR	progressive	NR
Gonzalez et al., (2014)	Body Weight/Machines	2	6	3	8 to 15	NR	NR	NR
Hewitt et al., 2018	Pneumatic/Balance	2	25	2 to 3	10 to 15	NR	Moderate (CR10)	60
Marques et al., (2011)	Machines	3	32	2	6 to 8	≥2 min	75–85% 1RM	60
Martins et al. (2011)	Calesthetics/Elastic Bands	3	16	varied	varied	3 min	progressive	45
Nicklas et al., (2016)	Machines	3	20	3	10	±1 min	70% 1RM	NR
Ramirez-Campillo et al., (2016)	High-speed RT	2	12	3	varied	±1 min	75% 1RM	50 to 70
Shiotsu & Yanagita, (2018)	Machines	2	10	3	8 to 12	NR	60–70% 1RM	NR
Sparrow et al., (2011)	Body Weight/Free Weights	3	26	2	12	NR	varied	60
Sylliaas et al., (2011)	Body Weight/Machines	3	12	3	8 to 12	NR	80% 1RM	45 to 60

NR = not reported, progressive = article only stated progressive resistance training when referring to the load applied, 1RM = 1 repetition maximum, CR10 = Borg rating of perceived exertion CR10, MTL = maximum tolerated load.

**Table 4 ijerph-18-01759-t004:** Reported dropouts & adverse events.

Authors	Drop-outs	Explanation	Adverse Events	Explanation
Alfieri et al., (2012)	5	1 ankle fracture, 1 rib fracture, 1 uncontrolled HF, 1 knee pain, 1 gave up	NR	
Cancela Carral et al., (2019)	2	death	NR	
Fahlman et al., (2011)	4	NR	NR	
Gonzalez et al., (2014)	0		0	
Hewitt et al., 2018	16	15 deceased, 1 moved away	4	3 musculoskeletal aches/pains, 1 noninjurious fall.
Marques et al., (2011)	8	Medical issues unrelated (*n* = 3) Disinterest (*n* = 3) Personal reasons (*n* = 2)	0	
Martins et al. (2011)	0		NR	
Nicklas et al., (2016)	7	3 personal health issues, 2 caretaking, 1 changed mind, 1 lost to follow-up	2	2 musculoskeletal complaints
Ramirez-Campillo et al., (2016)	0		0	
Shiotsu & Yanagita, (2018)	3	3 private reasons	NR	
Sparrow et al., (2011)	3	NR	8	8 musculoskeletal
Sylliaas et al., (2011)	5	1 nursing home, 1 died, 3 illness	NR	

NR = not reported.

**Table 5 ijerph-18-01759-t005:** Study conclusions.

Authors	Variable	Tools	Conclusion
Alfieri et al., (2012)	Balance	Timed Up and Go (TUG); Berg; Oscillation of the body’s center of pressure	Both multisensory and RT interventions improved static and dynamic mobility in healthy elderly subjects.
Cancela Carral et al., (2019)	Balance	TUG	Muscle strength intervention programs may help promote healthy lifestyles by maintaining autonomy, improving function, and balance.
Fahlman et al., (2011)	Gait	Velocity (m/min), step time (seconds), step length (cm): GAITRite mat	Eight weeks of RT increased the parameters of velocity and step length. Additional emphasis on gait training could improve gains even further.
Gonzalez et al., (2014)	Balance	Single leg balance	These findings support the use of progressive resistance training for untrained older adults to improve balance.
Hewitt et al., 2018	Gait & Balance	Short Physical Performance Battery (SPPB)	Moderate-intensity PRT and high-level balance exercise significantly reduced falls and improved SPPB performance.
Marques et al., (2011)	Balance	8-foot Up and Go (8-ft UG)	8-month RT, but not AT, can induce significant bone adaptation in older women and both regimens elicited significant gains in balance.
Martins et al., (2011)	Balance	8-ft UG	Both AT and RT interventions improved functional fitness.
Nicklas et al., (2016)	Gait & Balance	gait speed; SPPB; chair rise	Both RT and RT + Calorie Restriction groups increased in gait speed, SPPB score, and chair rise time.
Ramirez-Campillo et al., (2016)	Balance	8-ft UG; Bilateral balance w/Bertec BP5050 balance plate platform	2 or 3 training sessions/week of RT (equated for volume and intensity) are equally effective for improving physical performance and quality of life of older women.
Shiotsu & Yanagita, (2018)	Gait & Balance	10-m walk speed; TUG; single-leg balance with eyes open; Functional Reach Test (FRT)	10-m walk speed significantly increased in all training groups; Combined AT & moderate-intensity RT resulted in significant improvements in dynamic balance capacity.
Sparrow et al., (2011)	Balance	Single leg balance (eyes open) and Tandem stance	A home-based RT program for older adults resulted in significant improvements in muscular strength and balance.
Sylliaas et al., (2011)	Gait & Balance	Berg; TUG; 10-m walk speed	Significant improvements in BBS, sit-to-stand, TUG, and 10 m walk speed.

## Data Availability

Not applicable.

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
