# Peer review of "Influence of Resistance Training on Gait & Balance Parameters in Older Adults: A Systematic Review"

_ijerph, 2021, doi:10.3390/ijerph18041759_

Round 1

Reviewer 1 Report

Generally, this Review is interesting and easy to follow. 

I have the general recommendation to review the English language of the Discussion. 

Minor coments:

L178. I believe the studies are twelwe and not eleven.

L231 and in whole Discussion. I suggest using the past tense (e.g., "suggested" here) when referring to results from a study. 

L247. Please, specify what "improvement" means in this context. 

L252. "RT" already includes "training". 

L270. "in increasing".

L271. I would suggest commenting briefly on the use of other alternative tests to 1RM (e.g., 3RM, 5RM or prediction equation) used in this population.

Author Response

Reviewer 1

L178. I believe the studies are twelve and not eleven.

Yes, that has been corrected. Thanks!

L231 and in whole Discussion. I suggest using the past tense (e.g., "suggested" here) when referring to results from a study. 

Yes, we agree that when talking about a past study we use the past tense when not in direct relation to the current review. The corresponding author (a native English speaker) has reviewed the discussion once again and corrected any errors that may have occurred.

L247. Please, specify what "improvement" means in this context. 

The authors have changed the word improve to enhance just to be clear. Since we are talking about various aspects of gait, it can be hard to define all “improvements” (enhancements, developments, reinforcement, strengthening… etc.) that may take place due to resistance training.

L252. "RT" already includes "training". 

              We have deleted the extra “training”.

L270. "in increasing".

              Corrected

L271. I would suggest commenting briefly on the use of other alternative tests to 1RM (e.g., 3RM, 5RM or prediction equation) used in this population.

Great suggestion… we have included further mechanisms to use in older populations with a reference. We have added; “The intense efforts of a 1-RM may produce unnecessary musculoskeletal loading that may not be recommended for certain populations such as the elderly. For this population, an alternative method would be to include one of the many 1-RM prediction equations which have been shown to be a good predictor of an individual’s true 1-RM.”

Reviewer 2 Report

Main comments

General remarks to the authors

The importance of exercise in aging is clear and fundamental to public health. The theme is timely and important, and my concerns are not major, thus, I’m sure that the authors will reply successfully. I suggest including one possible rationale joining the strength capacity to functional mobility and balance below.

Minor points:

Lines 23, 24 – consider using “self-selected walking speed” instead of "straight-line walking speed"

Line 26 – daily living activities instead of activities of daily living.

Line 58 – Sarcopenia is defined as a syndrome characterized by an involuntary loss skeletal strength and mass. Please adjust.

Lines 66-68 – Consider using these references

Gomeñuka, N. A., Oliveira, H. B., Silva, E. S., Costa, R. R., Kanitz, A. C., Liedtke, G. V., ... & Peyré-Tartaruga, L. A. (2019). Effects of Nordic walking training on quality of life, balance and functional mobility in elderly: A randomized clinical trial. PloS one, 14(1), e0211472.

 to support the following sentence:

Traditionally, aerobic training programs have been used to reverse the effects of the 66 above-mentioned pathologies as well as an improvement in the health status of the elderly.

Lines 78-89 – Even interesting, your background miss gait and functional mobility points. I’d suggest including the following rationale: The reduced SSWS and lower walking metabolic economy may be related to dynapenia from aging process. The mechanistic link seems to be increased muscle co-contraction from lower limb muscles (Mian et al., 2006; Gomeñuka et al., 2020). Even the balance may be included in this rationale because the higher level of muscle co-contraction is a product of joint instability (due, in turn, to dynapenia).

Gomeñuka, N. A., Oliveira, H. B., da Silva, E. S., Passos-Monteiro, E., da Rosa, R. G., Carvalho, A. R., ... & Peyré-Tartaruga, L. A. (2020). Nordic walking training in elderly, a randomized clinical trial. Part II: Biomechanical and metabolic adaptations. Sports medicine-open, 6(1), 1-19.

Mian, O. S., Thom, J. M., Ardigò, L. P., Narici, M. V., & Minetti, A. E. (2006). Metabolic cost, mechanical work, and efficiency during walking in young and older men. Acta physiologica, 186(2), 127-139.

Table 1  - Ramirez-Campillo et al. (2016)

Table 2 – Define Vets and Spouse, NR

  • Idem to remaining tables. All tables and figures need to be self-explanatory.

  • Where are the table legends?

Line 197 – deleting ‘for time’, and includes table 5 inside of sentence (previously of period).

Line 200 – how many studies found positive adaptations on balance?

Lines 214 -218 – sentiment? With all respect to classic material from Whittle, this sentence is completely out of scope for your paper and, indeed, it contradicts the current trend to reinforce the importance of gait analysis in the context of global health in aging. You may want to state that tests with more complex situations (double tasks, negotiation of obstacles, etc.) may indicate specific adaptations, but it is still speculation, and is outside the scope of the study.

  • I think you can concluding that more studies comparing types of resistance training are needed to evaluate possible gains from resistance training in details.
  • I miss important systematic review similar to your study:
  •  
  • Olsen, P. Ø., Termannsen, A. D., Bramming, M., Tully, M. A., & Caserotti, P. (2019). Effects of resistance training on self-reported disability in older adults with functional limitations or disability–a systematic review and meta-analysis. European Review of Aging and Physical Activity, 16(1), 1-25.

Lacroix, A., Hortobagyi, T., Beurskens, R., & Granacher, U. (2017). Effects of supervised vs. unsupervised training programs on balance and muscle strength in older adults: a systematic review and meta-analysis. Sports medicine, 47(11), 2341-2361.

Jadczak, A. D., Makwana, N., Luscombe-Marsh, N., Visvanathan, R., & Schultz, T. J. (2018). Effectiveness of exercise interventions on physical function in community-dwelling frail older people: an umbrella review of systematic reviews. JBI Evidence Synthesis, 16(3), 752-775.

  • Consider using self-selected walking speed (SSWS) than "straight-line walking speed" , and sometimes “functional mobility” may be advisable.

Author Response

Reviewer 2

Lines 23, 24 – consider using “self-selected walking speed” instead of "straight-line walking speed"

Self-selected walking speed (comfortable walking speed) was not used in the studies reviewed. Walking speed for time (velocity of gait-maximal walking speed) was the most common primary outcome for the measure of gait (lines 208-211 explains).

Line 26 – daily living activities instead of activities of daily living.

With all due respect, Activities of Daily Living (ADL) is the proper terminology.

The activities of daily living (ADLs) is a term used to collectively describe fundamental skills that are required to independently care for oneself such as eating, bathing, and mobility.

Line 58 – Sarcopenia is defined as a syndrome characterized by an involuntary loss skeletal strength and mass. Please adjust.

The definition of Sarcopenia is not very clear as stated in the cited article in our review. They state: “Definition 1: having a low lean mass and a low grip strength. Definition 2: having a low lean mass, low muscle strength and a slow gait speed.” However, they go fewer and conclude that the definitions need to be further analyzed as it could include a fall risk component that is not yet identified.

We believe there is no effective difference and therefore justification to change between “involuntary loss of skeletal strength and mass” and what we have included as a  “progressive loss of strength and muscle mass…”

Lines 66-68 – Consider using these references

Gomeñuka, N. A., Oliveira, H. B., Silva, E. S., Costa, R. R., Kanitz, A. C., Liedtke, G. V., ... & Peyré-Tartaruga, L. A. (2019). Effects of Nordic walking training on quality of life, balance, and functional mobility in elderly: A randomized clinical trial. PloS one, 14(1), e0211472.

 to support the following sentence:

Traditionally, aerobic training programs have been used to reverse the effects of the 66 above-mentioned pathologies as well as an improvement in the health status of the elderly.

Thank you! That is a great addition and has been included. See line 68

Lines 78-89 – Even interesting, your background miss gait and functional mobility points. I’d suggest including the following rationale: The reduced SSWS and lower walking metabolic economy may be related to dynapenia from aging process. The mechanistic link seems to be increased muscle co-contraction from lower limb muscles (Mian et al., 2006; Gomeñuka et al., 2020). Even the balance may be included in this rationale because the higher level of muscle co-contraction is a product of joint instability (due, in turn, to dynapenia).

 Gomeñuka, N. A., Oliveira, H. B., da Silva, E. S., Passos-Monteiro, E., da Rosa, R. G., Carvalho, A. R., ... & Peyré-Tartaruga, L. A. (2020). Nordic walking training in elderly, a randomized clinical trial. Part II: Biomechanical and metabolic adaptations. Sports medicine-open, 6(1), 1-19.

Mian, O. S., Thom, J. M., Ardigò, L. P., Narici, M. V., & Minetti, A. E. (2006). Metabolic cost, mechanical work, and efficiency during walking in young and older men. Acta physiologica, 186(2), 127-139.

The authors are uncertain as to how that information would be applicable within the context of this review although it is an interesting point.

Table 1  - Ramirez-Campillo et al. (2016)

              Updated

Table 2 – Define Vets and Spouse, NR

              Updated

Idem to remaining tables. All tables and figures need to be self-explanatory.

We have further defined some of the tables with a more specific title, but could you please provide us with specific examples of the parts that are not self-explanatory? Thank you!

Where are the table legends?

We have included certain table legends for ease of reading, but we have not included legends for all tables.

Line 197 – deleting ‘for time’ and includes table 5 inside of sentence (previously of period).

The “for time” identification is necessary as mentioned above because the studies reviewed included a walking speed test for time (maximal walking speed) and not a comfortable walking speed test as the reviewer has mentioned.

We have corrected the period to proceed the parenthesis.

Line 200 – how many studies found positive adaptations on balance?

As stated in the text “All studies…” found positive adaptations for balance measures. However, to clarify further we have included “All eleven studies…”.

Lines 214 -218 – sentiment? With all respect to classic material from Whittle, this sentence is completely out of scope for your paper and, indeed, it contradicts the current trend to reinforce the importance of gait analysis in the context of global health in aging. You may want to state that tests with more complex situations (double tasks, negotiation of obstacles, etc.) may indicate specific adaptations, but it is still speculation, and is outside the scope of the study.

              Sentiment – “a view of or attitude toward a situation or event; an opinion”

We believe that this citation is adequate and useful in this context. Our review has found that only 5 studies measured gait parameters and all those gait parameters were measured in a unidirectional method, which is not entirely applicable to real life scenarios where obstacles and double tasks are often the cause of these devastating falls. At this point in the systematic review (discussion) we are trying to point out that other factors need to be further explored (as Whittle has stated) as not one of the studies included in this review looked at multidirectional or double task methods.

For that reason, we have added the statement below;

“Although the authors of this review believe that unidirectional gait assessment is an essential measurement, we also suggest that further research needs to include multidirectional and/or double task scenarios in order to better understand their utility in analyzing gait in older adults.”

I think you can concluding that more studies comparing types of resistance training are needed to evaluate possible gains from resistance training in details.

Although not stated in the conclusion, we do state in line 303-304; “Future studies need to be carried out to focus on the RT variable/s that allow for superior improvements in gait and balance.”

I miss important systematic review similar to your study:

Thanks to the reviewer for this comment. The authors have revised those previous papers and two of the three have been incorporated into our paper, by making stronger our findings (their results are in line with those reported by the current review). See further comments below for the specific details on each review.

Olsen, P. Ø., Termannsen, A. D., Bramming, M., Tully, M. A., & Caserotti, P. (2019). Effects of resistance training on self-reported disability in older adults with functional limitations or disability–a systematic review and meta-analysis. European Review of Aging and Physical Activity, 16(1), 1-25.

Although this a very interesting review and meta-analysis it reviews the effects of RT on Self-Reported Disability (not actual gait and balance variables) in patients that already demonstrate functional limitation and/or disabilities.

Lacroix, A., Hortobagyi, T., Beurskens, R., & Granacher, U. (2017). Effects of supervised vs. unsupervised training programs on balance and muscle strength in older adults: a systematic review and meta-analysis. Sports medicine, 47(11), 2341-2361.

This article is very supportive of our review, and we believe that it falls in line with our comments in the discussion. Therefore, we have included the following comment and citation of this review in line 306-310:

“However, it is noteworthy to report that a recent systematic review looking at the effects of supervised vs. unsupervised training programs on balance and muscle strength in older adults suggests that supervised training improved measures of balance and muscle strength/power to a greater extent than that of unsupervised programs.”

Jadczak, A. D., Makwana, N., Luscombe-Marsh, N., Visvanathan, R., & Schultz, T. J. (2018). Effectiveness of exercise interventions on physical function in community-dwelling frail older people: an umbrella review of systematic reviews. JBI Evidence Synthesis, 16(3), 752-775.

This is a very noteworthy umbrella review of reviews that looked at various exercise interventions (not simply RT) on various physical functions (not specifically balance and gait parameters) in frail older adults. However, the authors believe this article is very supportive of our review, and we believe that it confirms our comments in line 81 and for that reason we have included it as a reference and citation. See line 81.

Consider using self-selected walking speed (SSWS) than "straight-line walking speed”, and sometimes “functional mobility” may be advisable.

              See above comments on this topic.

Round 2

Reviewer 2 Report

INFLUENCE OF RESISTANCE TRAINING ON GAIT & BALANCE PARAME TERS IN OLDER ADULTS: A SYSTEMATIC REVIEW

Dear authors,

You have done a good job, and all questions were replied successfully.